# Genome-Wide Identification and Expression Profiling of Monosaccharide Transporter Genes Associated with High Harvest Index Values in Rapeseed (*Brassica napus* L.)

**DOI:** 10.3390/genes11060653

**Published:** 2020-06-15

**Authors:** Liyuan Zhang, Chao Zhang, Bo Yang, Zhongchun Xiao, Jinqi Ma, Jingsen Liu, Hongju Jian, Cunmin Qu, Kun Lu, Jiana Li

**Affiliations:** 1Chongqing Rapeseed Engineering Research Center, College of Agronomy and Biotechnology, Southwest University, Chongqing 400715, China; Liyuanzhang0215@163.com (L.Z.); 18083606406@163.com (C.Z.); sheepneck@hotmail.com (B.Y.); xzc729609100@163.com (Z.X.); jinqima1996@163.com (J.M.); Ljs282355@email.swu.edu.cn (J.L.); hjjian518@swu.edu.cn (H.J.); drqucunmin@swu.edu.cn (C.Q.); 2Academy of Agricultural Sciences, Southwest University, Chongqing 400715, China; 3State Cultivation Base of Crop Stress Biology for Southern Mountainous Land of Southwest University, Beibei, Chongqing 400715, China; drlukun@swu.edu.cn

**Keywords:** monosaccharide transporter, phylogenetic analysis, expression pattern, harvest index, *Brassica napus*

## Abstract

Sugars are important throughout a plant’s lifecycle. Monosaccharide transporters (MST) are essential sugar transporters that have been identified in many plants, but little is known about the evolution or functions of MST genes in rapeseed (*Brassica napus*). In this study, we identified 175 MST genes in *B. napus*, 87 in *Brassica oleracea*, and 83 in *Brassica rapa*. These genes were separated into the sugar transport protein (STP), polyol transporter (PLT), vacuolar glucose transporter (VGT), tonoplast monosaccharide transporter (TMT), inositol transporter (INT), plastidic glucose transporter (pGlcT), and ERD6-like subfamilies, respectively. Phylogenetic and syntenic analysis indicated that gene redundancy and gene elimination have commonly occurred in *Brassica* species during polyploidization. Changes in exon-intron structures during evolution likely resulted in the differences in coding regions, expression patterns, and functions seen among *BnMST* genes. In total, 31 differentially expressed genes (DEGs) were identified through RNA-seq among materials with high and low harvest index (HI) values, which were divided into two categories based on the qRT-PCR results, expressed more highly in source or sink organs. We finally identified four genes, including *BnSTP5*, *BnSTP13*, *BnPLT5*, and *BnERD6-like14*, which might be involved in monosaccharide uptake or unloading and further affect the HI of rapeseed. These findings provide fundamental information about MST genes in *Brassica* and reveal the importance of *BnMST* genes to high HI in *B. napus.*

## 1. Introduction

Photosynthetic products play essential roles in plant growth and development. These products, particularly sugars (including polyols, monosaccharides, and sucrose), are synthesized in photosynthetic organs (source tissues) [1,2] and transported to heterotrophic cells (sink tissues) [3]. The transport and distribution of sugars are important for maintaining the balance between source and sink tissues [4,5]. Plants contain two major types of sugar transporters: sucrose transporters and monosaccharide transporters (MSTs) [6]. *MSTs* are important transmembrane transporters that have been identified in many land plants and that function in carbohydrate flux [7,8].

The transport and distribution of sugars in plants occur via a complex process. Most plants synthesize starch in source organs following photosynthesis during the daytime [1]. At night, the starch is broken down via a hydrolytic or phosphorolytic pathway [9,10], allowing the products to be easily transported. Sucrose is the major form of photoassimilate that is transported from the source to sink organ via the phloem [3,11]. Following the long-distance transport of sucrose through the phloem, sucrose unloading into the sink tissue occurs via two major processes: symplasmic unloading [12,13,14] and apoplasmic unloading [10]. During symplasmic unloading, sucrose is directly exported into sink tissue. In contrast, both an enzyme (such as the cell wall-bound invertase (*CWI*)) [15] and a monosaccharide transmembrane transporter (MST) are required for apoplasmic unloading. During this process, the sucrose is broken down into different monosaccharides by invertase [14,16,17], and the monosaccharides are then transported by *MSTs* [18,19].

The first monosaccharide transporter gene identified, *CkHUP1*, was cloned via differential screening in *Chlorella*. This gene encodes a monosaccharide-H^+^ symporter [20]. Based on this finding, *MST* genes highly homologous to *CkHUP1* were cloned in *Arabidopsis thaliana*, as well as tobacco [21,22]. Increasing numbers of *MST* genes have been identified in various plants, including 53 *MSTs* in *Arabidopsis* [8], 64 in rice [23], and 5 in grapevine [24]. *MSTs* are important members of the major facilitator superfamily [25,26], and all of them contain 12 transmembrane domains. In *Arabidopsis*, *MSTs* can alter cellular sugar partitioning and sugar signaling and further increase the seed yield and biomass [27]. In rice, *OsMST4* functions to transport monosaccharide during the seed developmental process [28]. Ngampanya′s study provided a new insight into *OsMST5*, which functions in pollen development [29]. However, to date, no *MSTs* have been identified in *Brassica napus*.

Rapeseed (*Brassica napus* L.; AACC, 2n = 38) is the most important oilseed crop worldwide. Its seeds can be used to produce oil, its stem and leaves can be used for animal feed, and it can also be used to produce both biofuel and healthcare products [30]. The harvest index (HI), i.e., the ratio of seed yield to aboveground biomass, is one of the most important phenotypic traits of rapeseed. In the past few decades, rapeseed yields have significantly improved, but its HI remains lower than that of other crops [31,32]. Among the three factors affecting the HI—“source”, “flow”, and “sink”—the lack of sufficient sources and large sinks are not the main factors limiting the HI in *B*. *napa* [33]. Instead, the process of “flow” is the key limiting factor for the seed yield [34,35]. *MST* genes have crucial effects on carbohydrate flux [7], thereby influencing the biomass and seed yields [28,36,37,38]. To date, *MSTs* have been reported in various plant species, including *Arabidopsis* [8,23], grape [39], rice [38,40], cassava [41], pear [42], and *Lupinus polyphyllus* [43]. However, a genome-wide identification and expression analysis of *MST* genes has not previously been performed in *B*. *napus*.

In the present study, to explore the evolutionary relationships of *MST* genes in *Brassica* species, we performed genome-wide identification of *MSTs* in *B*. *napus* and its two parental species *B*. *oleracea* and *B*. *rapa*. We then performed phylogenetic analysis of these genes among three species, as well as *A. thaliana*. In total, we identified 175 MST genes in *B*. *napus*. By examining RNA sequencing (RNA-seq) data from four HI *B*. *napus* materials, we identified 31 differentially expressed genes (DEGs) with significantly different expression levels between seeds and silique pericarps. The expression patterns of these 31 DEGs were confirmed by qRT-PCR in two materials with extremely high and low HI values. We ultimately identified four candidate genes that might promote carbohydrate flux and increase the HI in *B*. *napus*.

## 2. Materials and Methods 

### 2.1. Identification of MSTs in B. napus, B. rapa and B. oleracea

The deduced protein sequences of *AtMST* genes were obtained from the *Arabidopsis* Information Resource database (https://www.arabidopsis.org/index.jsp). The BnMSTs, BoMSTs, and BrMSTs were identified via two methods. First, BLASTp analysis [44] was performed against the *B. napus* genome database (BRAD, http://brassicadb.org/brad/index.php) [45] using the AtMST protein sequences as queries. The HMMsearch program (HMMER 3.0, http://hmmer.org/) was then used to further confirm the presence of protein domains using the AtMST Pfam numbers as queries.

### 2.2. Multiple Sequence Alignment and Phylogenetic Analysis

Multiple sequence alignment and phylogenetic analysis were performed to investigate the evolutionary relationships of the MSTs among the four species. Multiple sequence alignment was performed with MEGA 7.0 (Molecular Evolutionary Genetics Analysis) with default parameters. A phylogenetic tree was constructed using the neighbor-joining (NJ) method, with bootstrap analysis of 1000 replicates [46]. The tree was further visualized with FigureTree software.

### 2.3. Analysis of the Chromosomal Locations, Gene Structures, and Conserved Motifs of the BnMST Genes 

Detailed information about the locations of the *BnMST* genes, their deduced protein sequences, and the relationships between *Brassica* and *A. thaliana* genes was obtained from the *Brassica napus* Genome Database. The chromosomal distribution of the genes was visualized with Map-Chart2.2 [47]. The gene structures were mapped using the Gene Structure Display Server (GSDS) [48]. The conserved motifs were analyzed using MEME online (http://meme-suite.org/tools/meme) [49,50]; all parameters were set to default settings, except the maximum number of predicted motifs, which was set to 20.

In addition, we calculated the molecular weights (Mw) and isoelectric points (PI) of the BnMSTs using the ExPASy proteomics server database (https://www.expasy.org/tools/) [51]. The subcellular localizations of the proteins were predicted with MultiLoc2 (https://abi-services.informatik.uni-tuebingen.de/multiloc2/webloc.cgi) [52].

### 2.4. RNA-seq Analysis 

Sugar transporters may influence the HI through increased seed yields. To identify the DEGs from the set of 175 *BnMST* genes identified in high HI materials, we download and analyzed RNA-seq data for *B. napus* from the NCBI database (ID Number SRP072900). We extracted the transcriptome data for the 175 *BnMST* genes and constructed a heatmap using R-Studio. The data were generated from seven tissues (stems, mature leaves, buds on the main branch, seeds from the main branch, seeds from lateral branches, silique pericarps from the main branch, and silique pericarps from lateral branches) from four materials with a high or low HI and biological yield (BY) [32].

### 2.5. Plant Materials 

To identify the expression patterns of the differentially expressed *BnMST* genes, which expressed differently for materials with a high and low HI, seeds from the materials CQ24 (SWU47, High HI) and CQ46 (Ning You 12, Low HI) were obtained from the Chongqing Rapeseed Technology Research Center, China. The plants were cultivated under field conditions in Chongqing for 2 years (2016 and 2017). Ten plants of rapeseed were planted in two experimental plots, and each plot contained five individual plants. After the mature stage, the two main characters, consisting of the dry weight of the above-ground biomass yield and seed yield per plant, were investigated. The harvest index was then calculated with the ratio of seed yield to above-ground biomass. Finally, we obtained the HI data of 10 individual plants from two replications respectively. The phenotypes of CQ24 and CQ46 are shown in Table 1. Samples of different tissues during various growth periods (stems at 0 and 30 days after flowering; leaves at the beginning of flowering; and flowers, buds, seeds, and silique pericarps at 7, 14, 21, 30, and 40 d after flowering) were collected from the two materials, immediately frozen in liquid nitrogen, and stored at −80 °C until use.

### 2.6. RNA Extraction and Validation of RNA-seq Data by qRT-PCR 

RNA was extracted from the samples using an RNeasy Extraction Kit (Invitrogen, Carlsbad, CA, USA) and employed as a template to produce cDNA with a Reverse Transcription Kit (TaKaRa Biotechnology, Dalian, China). As described by Qu [53], quantitative reverse-transcription PCR (qRT-PCR) was performed to identify the expression patterns of the DEGs identified by RNA-seq using specific primers designed with Premier 5.0 [54]. The primers, which were confirmed using BLAST online tools (BRAD, http://brassicadb.org/brad/blastPage.php), are shown in Appendix A. The *UBC21* gene was used as an endogenous reference gene [55]. The relative expression level of each gene was calculated using the 2^−△△Ct^ method [56]. The qRT-PCR was completed with three technical repetitions. The results were visualized with GraphPad Prism5.0 software [57,58].

## 3. Results

### 3.1. Identification and Phylogenetic Analysis of MSTs

In *Arabidopsis*, 53 *AtMST*s have been identified [8]. In this study, using the amino acid sequences of the 53 AtMSTs as queries, we identified 175 *BnMST*, 87 *BoMST*, and 83 *BrMST* genes by both BLASTP and HMM analyses (Appendix A). Notably, only 50 *AtMST* genes matched the *BnMST* genes, whereas the three remaining *AtMST* genes (*At3G05160*, *At1G08890* (*AtERD6-like subfamily*), and *At1G05030* (*AtpGlcT subfamily*)) lacked homologs in *B. napus* (Table 2). Detailed information about the *BnMST* genes is provided in Appendix A. The molecular weights (Mw) of the BnMST proteins ranged from 20.06 (BnERD6-like5-1) to 161.72 (BnpGlcT3), and their isoelectric points (PI) ranged from 4.93 (BnTMT3-3) to 10.46 (BnERD6-like3-5) (Appendix A).

To explore the evolutionary relationships of the MSTs, we performed a phylogenetic analysis. The 295 *MST* genes were divided into seven subfamilies in this phylogenetic tree. This result is consistent with previous findings in *Arabidopsis* [8]. We named the seven subfamilies based on the types of monosaccharides they transport [8]: the sugar transport protein (STP), vacuolar glucose transporter (VGT), tonoplast monosaccharide transporter (TMT), plastidic glucose transporter (pGlcT), polyol transporter (PLT), inositol transporter (INT) and ERD6-like subfamilies (Figure 1).

### 3.2. Chromosome Locations of BnMST Genes and Duplication Analysis 

The 175 *BnMST* genes are unevenly distributed on the *B. napus* chromosomes (Figure 2A), including 86 genes in the A subgenome and 89 in the C subgenome. Of these, 153 *BnMST* genes were identified on the 19 *B. napus* chromosomes, and the 22 remaining *BnMST* genes were mapped to pseudo-molecule chromosomes Ann and Cnn or to Unn (unknown). Chromosomes A01, A03, A05, A06, C01, and C05 each contain more than 10 *BnMST* genes, whereas chromosomes A02, A04, A10, C02, C04, C06, and C09 contain fewer than six *BnMST* genes (Figure 2A). The syntenic relationships of the 175 *BnMST* genes are indicated by lines in Figure 2B. Sixty-nine homologous pairs were identified among the 175 *BnMST* genes (Figure 2B). The syntenic relationships among the four species are shown in Figure 2C. The copy numbers of the *MST* genes in *A*. *thaliana* and *B. napus* ranged from one to eight (Table 2), indicating that some genes were lost or duplicated during evolution. For example, the syntenic homologs of *AtERD6-like15*, *AtERD6-like16*, and *AtpGlcT4* have been lost in *Brassica* species. However, three homologous syntenic genes of *AtERD6-like6* are present in *B. rapa* and *B. oleracea* and six are present in *B. napus* (three each in the A and C subgenomes) (Figure 2C, Table 2).

### 3.3. Exon-Intron Structures and Conserved Motif Analysis of the BnMSTs

We analyzed the exon-intron structures and conserved motifs of the BnMSTs to obtain additional information about their protein profiles (Figure 3). The number of exons ranged from 2 (*BnSTP4-2*) to 33 (*BnVGT1-1*). Genes in the BnERD6-like subfamily (except *BnERD6-like1*, *BnERD6-like3*, and *BnERD6-like6*) and BnSTP subfamily (*BnSTP2*, *BnSTP4*, *BnSTP6*, *BnSTP8*, *BnSTP9*, *BnSTP10*, and *BnSTP11*) contained significantly fewer exons than genes in the other subfamilies. This analysis, combined with phylogenetic analysis of the BnMSTs, indicated that the higher the homology of the sequences, the more similar their genetic structures. For example, *BnSTP9-1* and *BnSTP9-4* have highly similar gene structures, as do *BnSTP10-2* and *BnSTP10-6*, whereas the exon-intron structures of *BnSTP9-1, BnSTP9-4* vs. *BnSTP10-2, BnSTP10-6* are significantly different.

Twenty conserved motifs were predicted using the MEME server (Figure 4 and Appendix A). Motif 10 is present in all BnMST family members except STP, and motif 13 is in all BnMST family members except INT and PLT. Motifs 9 and 20 are only present in the BnSTP subfamily and motif 19 is only present in the BnSTP, BnINT, and BnERD6-like subfamilies. The BnSTP and BnpGlcT subfamilies contain unique motif 15 and 17, respectively. 

We predicted the subcellular localizations of the proteins using the online tool MultiLoc2 (Appendix A). Most of the BnMST proteins were predicted to be located in the cytoplasm (45 BnMST proteins), chloroplast (34 BnMST proteins), or nucleus (12 BnMST proteins), and the others were predicted to be in secretory pathways.

### 3.4. RNA-seq Analysis of the 175 BnMST Genes

To identify DEGs of the *BnMST* genes among materials with a high and low HI, we analyzed publicly available RNA-seq data (Appendix A). We then constructed a heatmap of the expression patterns of the 175 *BnMST* genes (Figure 5). The genes showed various expression patterns, depending on the subfamily classification. Overall, BnpGlcT subfamily genes were expressed at significantly higher levels than genes in the six other subfamilies, especially the BnERD6-like subfamily. However, many genes had different expression patterns, even within the same subfamily. For example, *BnPLT1*, *BnPLT2*, *BnPLT6*, *BnVGT1*, *BnTMT3*, *BnINT3*, *BnINT4*, *BnERD6-like1*, *BnERD6-like4*, *BnERD6-like7*, *BnERD6-like8*, *BnERD6-like12*, *BnERD6-like13*, *BnSTP10*, and *BnSTP11* were expressed at low levels in almost every organ (marked with yellow in Figure 5), but their homologous genes were expressed at high levels in specific tissues (indicated by blue arrows, green and red ellipses in the heatmap, Figure 5). Among these, many genes were differentially expressed between silique pericarps and seeds, including *BnPLT3*, *BnPLT4*, *BnPLT5*, *BnVGT1*, *BnVGT2*, *BnVGT3*, *BnTMT1*, *BnTMT2*, *BnINT1*, *BnINT2*, *BnpGlcT1*, *BnpGlcT2*, *BnERD6-like3*, *BnERD6-like5*, *BnERD6-like6*, *BnERD6-like9*, *BnERD6-like10*, *BnERD6-like14*, *BnSFP1*, *BnSFP2*, *BnSTP1*, *BnSTP3*, *BnSTP4*, *BnSTP5*, *BnSTP6*, *BnSTP7*, *BnSTP12*, and *BnSTP13*. We identified 31 genes that were specifically expressed in silique pericarps and seeds and were differentially expressed in materials with extremely high vs. low HI values (indicated by red ellipses in Figure 5).

### 3.5. qRT-PCR Analysis of DEGs in Different Tissues between Diverse Materials

We performed qRT-PCR to examine the expression patterns of the 31 DEGs in diverse tissues of plants at different stages of growth between materials with extremely high (CQ24) and low (CQ46) HI values (Figure 6, Table 1). The phenotype of plants and siliques between CQ24 and CQ46 are shown in Figure 6A,B, respectively. Phenotype analysis indicated that the traits of the harvest index (HI), seed yields (SY), and biological yields (BY) were significantly different between the two materials (Figure 6C).

These 31 genes were specifically expressed in different tissues, growth stages, and materials (Figure 7 and Appendix A). We divided the DEGs into two major categories based on the qRT-PCR results: *BnSTP1*, *BnSTP7*, *BnSTP13*, *BnTMT1*, *BnTMT2*, *BnINT2*, *BnpGlcT2*, *BnPLT5*, *BnERD6*, and *BnERD6-like5* were highly expressed in silique pericarps (Figure 7B), whereas *BnSTP5*, *BnSTP12*, *BnERD6*-*like10*, and *BnERD6*-*like14* were highly expressed in seeds (Figure 7A). In addition, *BnVGT2*-*2* was only expressed in line CQ24 (Appendix A). The expression levels of several DEGs increased rapidly in seeds and silique pericarps during the later period of plant growth and development (30-40 d after flowering), including *BnSTP5*, *BnSTP12*, *BnINT2*, *BnpGlcT1*, *BnpGlcT2*, *BnPLT3*, *BnPLT5*, *BnERD6-like5*, and *BnERD6-like10*. Perhaps the genes that are expressed at higher levels in silique pericarps (source organs) play roles in photosynthate loading into source organs, whereas genes expressed specifically in seeds (sink organs) might play important roles in the unloading of carbohydrates. These results suggest that these *BnMST* genes might increase the HI of *B. napus* by enhancing the translocation of assimilates to grains (seeds) and that the *BnMST* genes likely play essential roles in the transport of monosaccharides during the later stages of seed growth and development.

## 4. Discussion

*B. napus* (AACC, 2n = 38) is an allotetraploid plant derived from two diploid species (*B*. *rapa*, n = 10, and *B. oleracea*, n = 9) [59], as confirmed by numerous experimental crosses. The genetic relationships of these species are described by the “U′s triangle” model. Comparative genomic analysis between *A*. *thaliana* and *B*. *rapa* has clearly confirmed the occurrence of a whole genome triplication (WGT) event [60] millions of years ago [61,62]. In the current study, we identified 175 *MST* genes from *B*. *napus*. However, we identified one to eight homologs of each *AtMST* gene in *B*. *napus*, rather than six for every gene (Figure 2C, Table 2), perhaps due to genome shrinkage and redundancy. Indeed, Mun et al. reported that genome shrinkage and the differential loss of duplicated genes occurred after the WGT event [63], and other studies have provided insights into the processes of genome duplication, loss, or retention [64,65]. In addition, we identified 87 and 83 MST genes from *B*. *oleracea* and *B*. *rapa*, respectively. The number of *BnMST* genes in *B*. *napus* (175) is greater than the sum of *BoMST* and *BrMST* genes (Table 2), pointing to possible redundancy among the *BnMST* genes. When we constructed a phylogenetic tree for the four plant species, all *MST* genes were assigned to seven subfamilies (Figure 1) based on the nomenclature used in *A*. *thaliana* [8]. Interestingly, the seven subfamilies did not all follow the same patterns of gene redundancy. The INT subfamily was fully compliant with the WGT event, whereas the number of *B*. *napus* genes in the PLT, STP, and pGlcT subgroups was less than the sum of genes in *B*. *rapa* and *B*. *oleracea*, suggesting that gene elimination occurred in these three subgroup (Figure 2C, Table 2).

By combining the results of the gene structure, conserved motif, and protein analysis of the BnMSTs, we determined that genes from the same subgroup have similar features. For example, BnSTP and BnERD6-like subfamily members have significantly fewer exons than genes in the five other subfamilies (Figure 3). Similarly, only BnSTP subfamily members contain motif 9 and 20, and most BnMST family members (except BnSTP subfamily proteins) contain motif 10. Furthermore, motif 19 is only present in the BnSTP, BnERD6-like, and BnINT subfamilies (Figure 4). Interestingly, BnSTP (especially *BnSTP10*) and BnERD6-like subfamily members were expressed at much lower levels than members of the five other subgroups, as shown in the heatmap (Figure 5, genes marked with yellow triangles). In conclusion, gene redundancy and gene elimination occurred during the genomic evolution of *BnMST*, which further drove the diversification of homologous genes in *B*. *napus.* In addition, during the genomic evolution of the *BnMST* genes, changes in exon-intron structures might have resulted in different coding regions, thereby altering the expression patterns and functions of these genes [66].

Among *BnMST* family members, we were interested in identifying genes with vital functions in seed development and ripening. We therefore analyzed RNA-seq data for the phenotypic characters related to the HI (Appendix A) and used the gene expression patterns to construct a heatmap (Figure 5). Most, but not all, *BnMST* genes in the same subfamilies had similar expression patterns. For example, genes from the BnpGlcT subfamily were ubiquitously expressed in all tissues, whereas genes in the six other subfamilies were expressed in specific tissues (Figure 5). However, *BnSTP5* and *BnSTP13*, which belong to the same subfamily, showed completely different expression profiles: *BnSTP5* expression strongly increased during the later stage of seed development, whereas *BnSTP13* was primarily expressed in leaf tissue at the beginning of flowering and in stems at 30 d after flowering (DAF; Figure 7B). These results suggest that homology does not necessarily reflect a similar function. There were some discrepancies between the heatmap and qRT-PCR results, perhaps due to the different materials and diverse RNA samples examined.

Whereas the expression of *BnSTP5* strongly increased in seeds at 30 DAF, *BnSTP1* was mainly expressed in silique pericarps at 14–30 DAF (Figure 7A), suggesting that *STP1* might contribute to sugar uptake and is primarily expressed in the early stage of seed development [38]. *BnSTP13* was only strongly expressed in stems at 30 DAF and in leaves at the beginning of flowering. This expression pattern is in complete agreement with reports on *Arabidopsis*, which suggest that overexpressing *AtSTP13* could improve the glucose uptake capacity and increase the plant biomass or enhance disease resistance [36,37,67]. *STP8* is highly expressed in reproductive organs, whereas *STP7* plays an important role in sugar uptake and recycling in the cell wall [68]. *STP6* and *STP13* are the only genes known to function in fructose transport among STP subfamily members [67].

*AtPLT5* encodes another plasma membrane transporter responsible for fructose uptake; this gene is primarily expressed in sink tissues [69]. In contrast, in the current study, *BnPLT5* was primarily expressed in leaves at the beginning of flowering and in silique pericarps at 30–40 DAF (Figure 7B), perhaps because the silique pericarp can act as a source or sink tissue at different stages of plant development. The TMT subfamily members *TMT1* and *TMT2* are highly expressed in various tissues and might be important for the seed yield, whereas *TMT3* is barely expressed throughout a plant’s lifecycle [27,70]. The expression patterns of pGlct, INT, and VGT subfamily members were different from those identified in previous studies, perhaps due to the different species analyzed.

Interestingly, several genes were more strongly expressed in materials with low, as opposed to high, HI values, such as *BnSTP5*, *BnSTP12, BnERD6-like10*, and *BnERD6-like14* (Figure 7A). Among BnMST family members, only these genes were expressed at significantly higher levels in seeds than in silique pericarps. Luo et al. suggested that the HI is primarily influenced by the relationship between “source”, “flow”, and “sink” and that “flow” is the crucial limiting factor when the “source” is sufficient and the “sink” is not fully utilized [34]. Based on this finding, we hypothesize that *BnSTP5*, *BnSTP12,* and *BnERD6-like14* take part in the sugar unloading process in sink organs and that a negative feedback effect might occur among these *BnMST* genes when the “sink” is full. Perhaps the roles of these *BnMST* genes could be functionally verified in the future.

To better explore the localizations and interactions of the BnMST proteins, we predicted their subcellular localizations using an online tool. The 175 BnMST proteins are localized to diverse compartments, such as the cytoplasm, chloroplast, mitochondria, and nucleus (Appendix A). In *A*. *thaliana*, pGlcT and MEX1 function in the export of starch degradation products from chloroplasts [71]. However, we predicted that BnpGlcT1 is localized to chloroplasts, whereas BnpGlcT2 is localized to the cytoplasm (Appendix A), suggesting that functional segregation might occur in the same subfamily. Furthermore, BnSTP5, BnSTP12, and BnERD6-like14, which are encoded by genes with similar expression patterns, are all located in secretory pathways, indicating that some BnMST proteins likely interact with other proteins from different subfamilies.

## 5. Conclusions

In the current study, we performed a systematic study of the *BnMST* gene family. Genome-wide identification, phylogenetic analysis, and syntenic analysis among *B. napus*, *B. oleracea*, *B. rapa*, and *A. thaliana* indicated that gene redundancy and elimination occurred during the evolution of BnMST family members. Our analysis of RNA-seq data, gene structures, and conserved motifs indicated that changes in the exon-intron structure could lead to the presence of different coding regions and alter gene expression patterns and functions. Based on subcellular localization and expression analysis of DEGs between materials with two different HI values, *BnSTP5*, *BnSTP13*, *BnPLT5*, and *BnERD6-like14*, which are specifically expressed in seeds, silique pericarps, or stems, represent excellent candidate genes for further functional studies. Our findings provide basic information about *MST* genes in *Brassica napus* and uncover several candidate genes related to the HI for further analysis.

## Figures and Tables

**Figure 1 genes-11-00653-f001:**
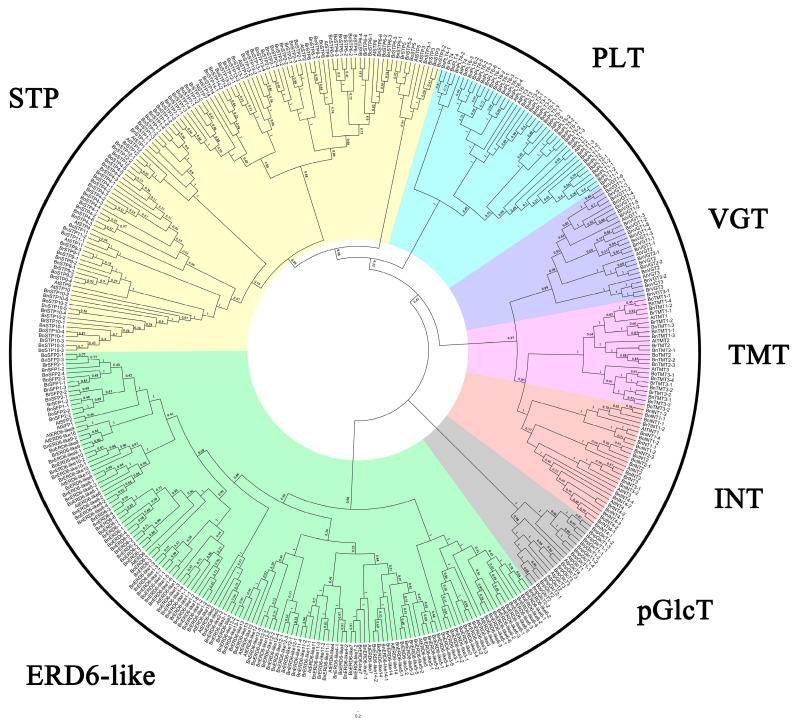
Neighbor-joining (NJ) tree of the monosaccharide transporter (MST) protein sequences from *Brassica napus*, *Brassica rapa*, *Brassica oleracea*, and *Arabidopsis thaliana*. All *MST* genes from the four species were divided into seven subfamilies and named PLT1 to STP14. In the names of the *BnMST*, *BoMST*, and *BrMST* genes, the first number indicates the congruent relationship with *Arabidopsis* and the second number represents the corresponding number in the respective species.

**Figure 2 genes-11-00653-f002:**
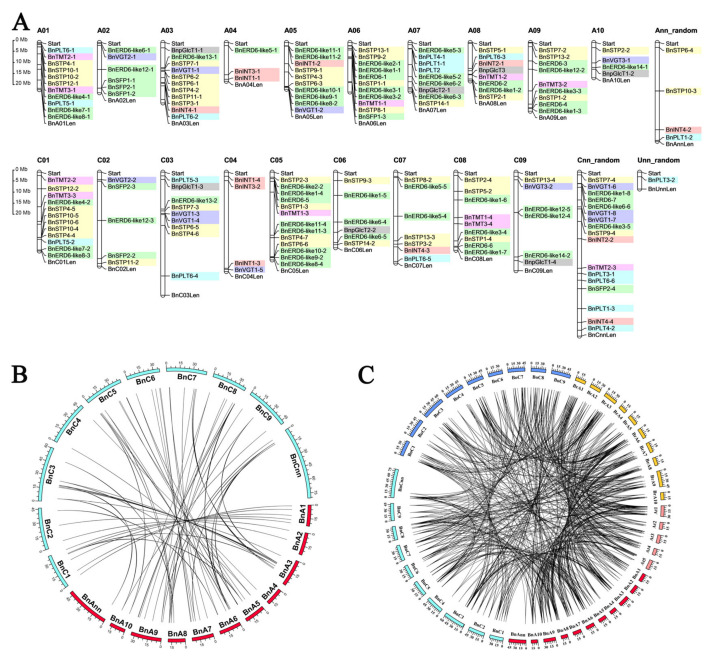
Chromosomal location of the *BnMST* genes in the *B*. *napus* genome and syntenic analysis of *MST* genes among *A*. *thaliana*, *B*. *napus*, *B*. *rapa*, and *B*. *oleracea*. (**A**) Chromosomal locations of the 175 *BnMST* genes. The length of each chromosome is marked by the words “Start” and “Len”. The seven background colors represent seven different subfamily members of BnMST respectively, as is the case for the colors shown in Figure 1. (**B**) Syntenic relationships of the 175 *BnMST* genes, as indicated by connecting lines. (**C**) Syntenic relationships of *MST* genes among the four species, as indicated by connecting lines. (**A**) and (**C**) are the two main chromosomes in *B*. *napus*; 01 to 10 represent chromosome numbers and Mb indicates megabases. Ann and Cnn represent pseudo-molecule chromosomes and Unn indicates that the specific position is unknown. The five different colors in Figure 2C means chromosomes from diverse species or the same species′ chromosome from diverse sub-genomes.

**Figure 3 genes-11-00653-f003:**
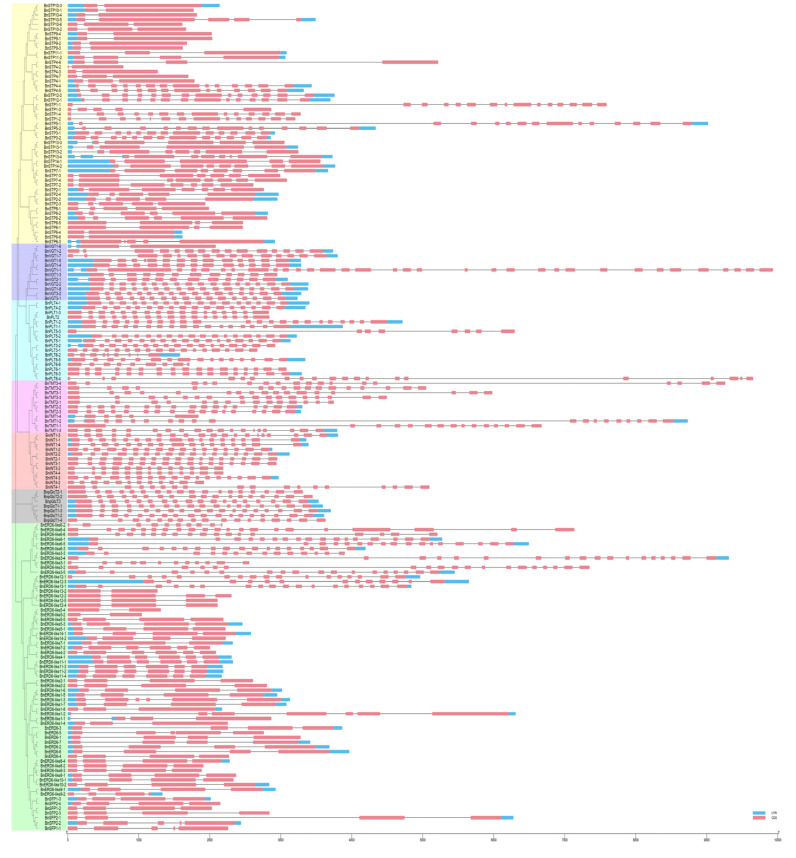
Exon-intron structures of the 175 *BnMST* genes based on their phylogenetic relationships. The phylogenetic tree was constructed using the protein sequences of 175 BnMSTs with 1000 bootstrap replicates.

**Figure 4 genes-11-00653-f004:**
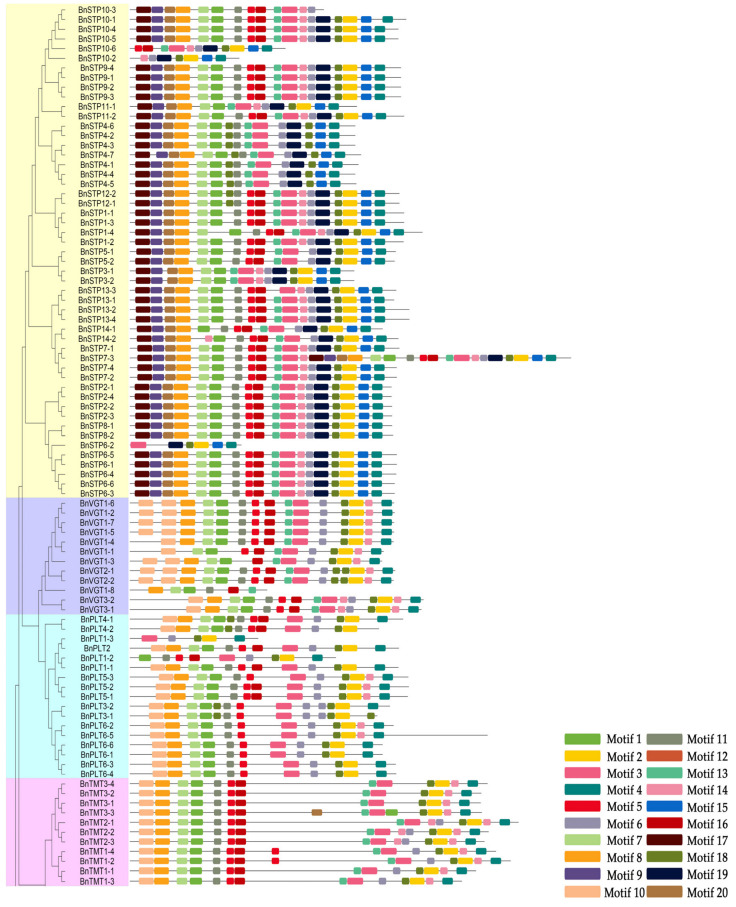
Conserved motifs of the 175 BnMSTs based on their phylogenetic relationships; gray lines indicate non-conserved sequences. Weblogo plots of the 20 conserved motifs are presented in Appendix A.

**Figure 5 genes-11-00653-f005:**
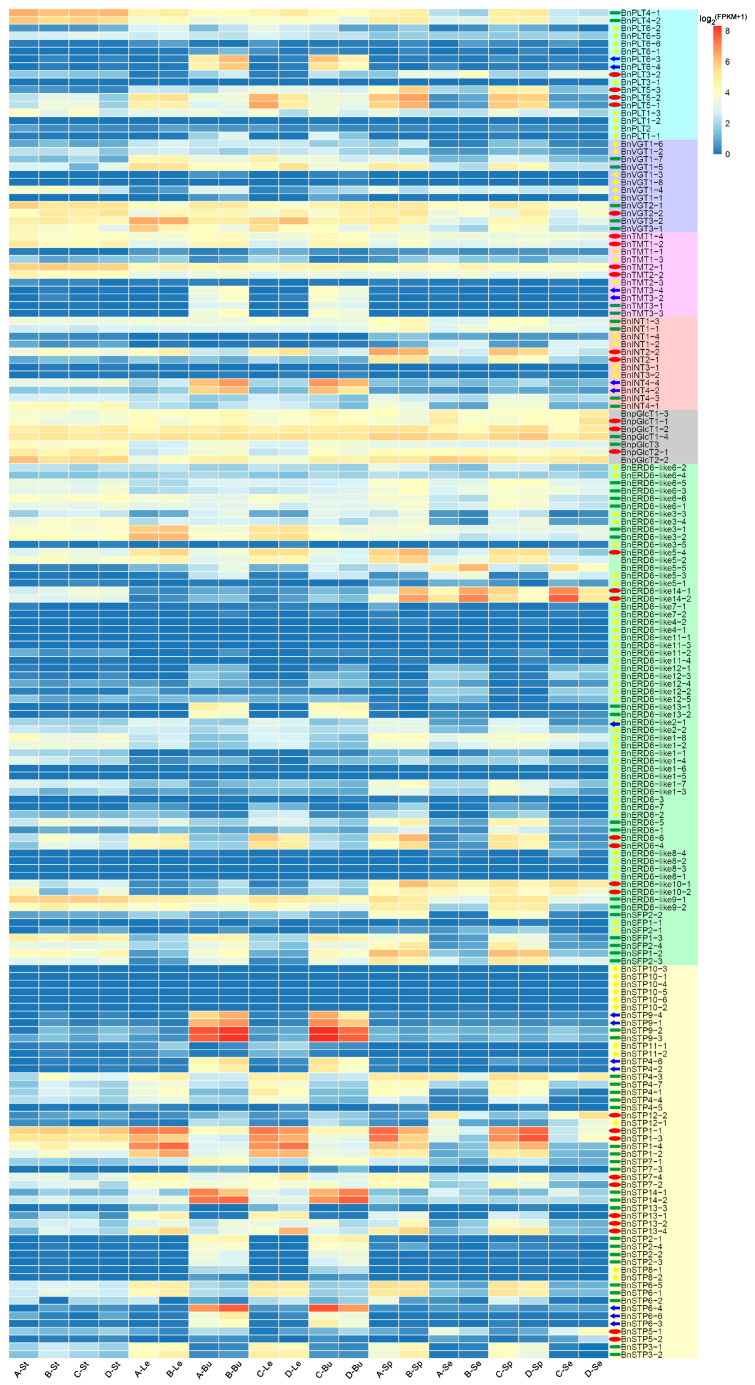
Heatmap of the expression patterns of the 175 *BnMST* genes in various tissues between different materials. A, B, C, and D represent different materials: A: High biological yield (BY) but low harvest index (HI); B: high BY and HI; C: low BY but high HI; D: low BY and low HI; J, Le, ZL, ZJ, CJ, ZS, and CS represent the different tissues: J: stems; Le: mature leaves; ZL: buds on the main branch; ZJ: silique pericarps from the main branch; CJ: silique pericarps from the lateral branch; ZS: seeds from the main branch; CS: seeds from the lateral branch. Different colors represent different expression levels.

**Figure 6 genes-11-00653-f006:**
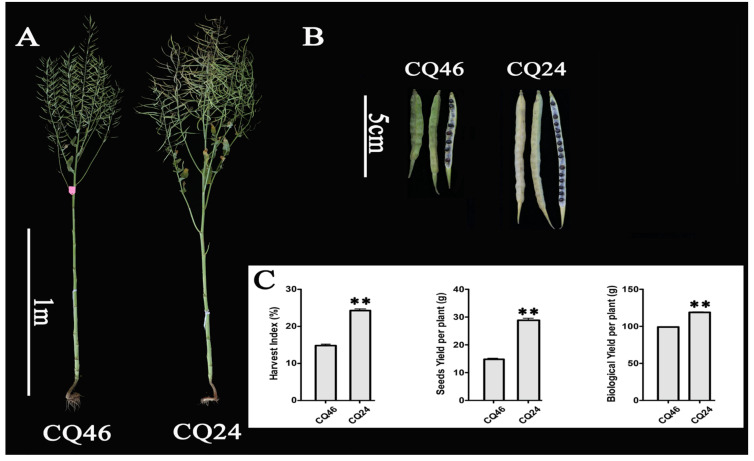
Phenotypic analysis of the materials CQ24 and CQ46 in *Brassica napus*. Phenotype of the plants and siliques is shown in (**A**) and (**B**), respectively. The traits of the harvest index (HI), seed yields (SY), and biological yields (BY) between the two materials are shown in (**C**). The value of the experiment was 2. CQ24 (SWU47) has a high HI, SY, and BY, whereas CQ46 (Ning You 12) has a low HI, SY, and BY. **: significant difference at *p* < 0.01.

**Figure 7 genes-11-00653-f007:**
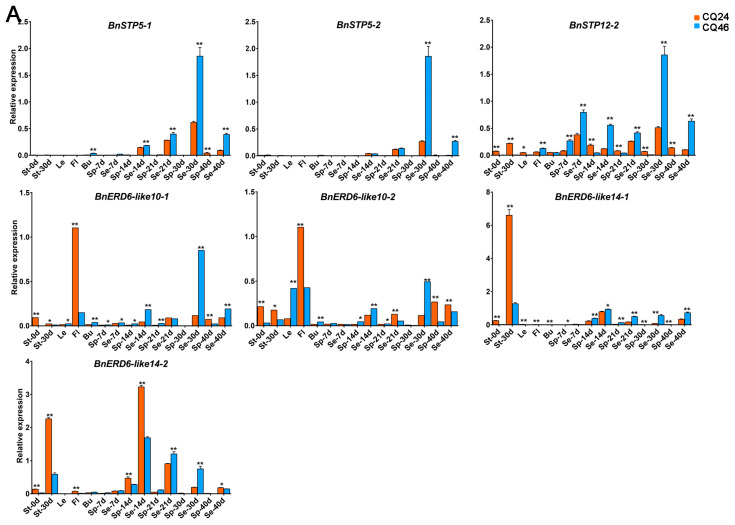
Expression patterns of the differentially expressed genes (DEGs) in different tissues and growth periods between two plant materials with an extremely high (CQ24) and low (CQ46) HI. (**A**) Expression patterns of 7 identified DEGs, which expressed extremely higher in source organs (seeds). (**B**) Expression patterns among 19 identified DEGs, which expressed extremely higher in sink organs (silique pericarp). The letter “d” indicates days after flowering; St: stem; Le: leaves at the beginning of flowering; Fl: flower; Bu: bud; Sp: silique pericarp; Se: seed; *: significant difference at *p* < 0.05; **: significant difference at *p* < 0.01.

**Table 1 genes-11-00653-t001:** Harvest index of plant materials during the two years of field tests.

Trait	Material	2016	2017	Mean Value	SEM	*P*-Value
Harvest Index /% (HI)	CQ24	24.75	24.33	24.28	0.47	0.004
	CQ46	15.21	14.43	14.82	0.39	
Seed Yield per Plant /g (SY)	CQ24	29.60	28.13	28.86	0.73	0.004
	CQ46	15.21	14.37	14.79	0.42	
Biological Yield per Plant /g (BY)	CQ24	119.60	118.15	118.87	0.73	0.002
	CQ46	98.60	99.55	99.08	0.48	

**Table 2 genes-11-00653-t002:** The number of homologous genes among the A and C sub-genomes of *Brassica napus*, *Brassica rapa*, *Brassica oleracea*, and their homologs in *Arabidopsis.*

Gene Name	*A. thaliana*	*B. rapa*	*B. oleracea*	*B. napus (A)*	*B. napus (C)*
**Zero-Copy in *B. napus***					
*ERD6-like15*	1	0	0	0	0
*ERD6-like16*	1	0	0	0	0
*pGlcT4*	1	0	0	0	0
**One-Copy in *B. napus***					
*PLT2*	1	0	1	1	0
*pGlcT3*	1	1	1	1	0
**Two-Copy in *B. napus***					
*ERD6-like2*	1	1	1	1	1
*ERD6-like4*	1	0	1	1	1
*ERD6-like7*	1	1	1	1	1
*ERD6-like9*	1	1	1	1	1
*ERD6-like10*	1	1	1	1	1
*ERD6-like13*	1	1	1	1	1
*ERD6-like14*	1	1	1	1	1
*pGlcT2*	1	1	1	1	1
*STP3*	1	1	1	1	1
*STP5*	1	1	1	1	1
*STP8*	1	1	1	1	1
*STP11*	1	0	1	1	1
*STP12*	1	1	1	1	1
*STP14*	1	1	1	1	1
*VGT2*	1	1	1	1	1
*VGT3*	1	1	1	1	1
*INT2*	1	1	1	1	1
*INT3*	1	1	0	1	1
*PLT3*	1	1	1	1	1
*PLT4*	1	0	0	1	1
**Three-Copy in *B. napus***					
*SFP1*	1	2	0	3	0
*TMT2*	1	1	1	1	2
*PLT1*	1	3	2	2	1
*PLT5*	1	2	2	1	2
**Four-Copy in *B. napus***					
*ERD6-like8*	1	1	2	2	2
*ERD6-like11*	1	2	0	2	2
*SFP2*	1	2	3	1	3
*INT1*	1	2	2	2	2
*INT4*	1	2	2	2	2
*pGlcT1*	1	2	2	2	2
*TMT1*	1	2	2	2	2
*TMT3*	1	2	2	2	2
*STP1*	1	2	2	2	2
*STP2*	1	2	2	2	2
*STP7*	1	2	2	2	2
*STP9*	1	2	2	2	2
*STP13*	1	2	2	2	2
**Five-Copy in *B. napus***					
*ERD6-like3*	1	2	1	3	2
*ERD6-like5*	1	2	2	3	3
*ERD6-like12*	1	2	2	2	3
**Six-Copy in *B. napus***					
*ERD6-like6*	1	3	3	3	3
*STP6*	1	4	3	4	3
*STP10*	1	3	4	3	3
*PLT6*	1	3	3	3	3
**Seven-Copy in *B. napus***					
*ERD6*	1	3	4	4	3
*STP4*	1	3	5	2	3
**Eight-Copy in *B. napus***					
*ERD6-like1*	1	3	3	4	4
*VGT1*	1	3	4	2	6
TOTAL	53	83	87	87	88

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
