# Peer review of "Genome-Wide Identification and Expression Profiling of Monosaccharide Transporter Genes Associated with High Harvest Index Values in Rapeseed (Brassica napus L.)"

_genes, 2020, doi:10.3390/genes11060653_

Round 1

Reviewer 1 Report

The manuscript of Zhang et al. describe a study of the Monosaccharide Transporter gene family in rapeseed using bioinformatics on the basis of publicly available data (MST and RNAseq databases). This provides extensive information on this complex gene family that could be used in order to improve yield on this important crop. Moreover, using RT-qPCR, the authors were able to select some gene candidates that could play an important role in this process through their involvement in sugar transport to sink structures like seeds.

The manuscript is well written and the methods are correctly described. The introduction gives the important informations necessary to understand the work. The presentation of the data is also of good quality except for figure 7 because of the quality of the image (at least in the version I reviewed). This figure needs to be improved in terms of quality or needs to be reduced. The conclusions are in adequation with the data and are interesting.

My only remark concerns the RT-qPCR data which are based on the use of a single reference gene whereas the authors had the possibility to use a second reference gene using the data provided in bibliographic reference n°58. But this is not a main concern for publication in Genes.

This work is not particularly original in its approach or topic but provides interesting data for the rapeseed community.

In conclusion, I recommand this manuscript for publication in Genes with minor modification of figure n°7.

Author Response

Dear Reviewer:

Thank you for the kind suggestions and comments, please see our responses in the attachment.

Reviewer 2 Report

Authors have studied a class of sugar transporters, monosaccharide transporters (MST) in rapeseed (Brassica napus). Little is known about the evolution or functions of MST in this crop. They have identified on this plant more than 173 gene copies and have  realized several gene expression analyses to identify those correlated with plant material phenotyped with high and low harvest index (HI) values. This approach permitted to identify 4 BnMST genes associated to high HI in B. napus.

This is an interesting work and analysis has been done in an accurate and robust way.

I may have few questions and suggestions to improve the manuscript.

Line 71. Authors said “MST genes have crucial effects on carbohydrate flux [7], thereby influencing biomass and seed yields [29, 38-40].”

However, authors have only quantified seed yields and they do not give any indication about biomass in those experiments. Could we expect a correlation between both parameters and an implication of MST on it?

Table 1. How harvest Index /%(HI) is calculated, which is value of n= ? in those experiments?

Figure 1. the bootstrapping values are not indicated on each branch of the tree.

Figure 2. A) Describe in legend what colors mean. C) Describe in legend what colors mean.

Figure 5. Different colors represent different expression levels = log2, log10, FPKM?

Figure 6. which is value of n= ? in those experiments?

Author Response

(The authors gave the same response as above.)

Reviewer 3 Report

Li-yuan Zhang  and co-workers have performed a study on MST genes in Brassica to determine genes responsible for high harvest index (HI) in Brassica napus. They estimated the influence of candidate genes by examining their expression in material from two types of plant, CQ24 (SWU47) and CQ46 (Ning You 12) with high and low HI, respectively.

The authors made a strong effort to use syntenic analysis  and RNA-seq analysis of BnMST genes for identifying genes involved in promote carbohydrate flux and increase HI in B. napus. I am not an expert in phylogenetic and transcriptomic analysis thus I cannot comment in detail on the quality of the study.

In many ways the work represents a resource study with limited mechanistic insight at this stage. It is understandable that extended functional gene characterizations are beyond the scope of the study; however, in my opinion, the manuscript would be more interesting if it had been expanded on the characterization of selected genes expressed in various level in CQ24 and CQ46 plants (using RNAi or CRISPR approach) as a successful outcome of the analysis. The phenotypes detected in two types of plants, CQ24 and CQ46 respectively, should be backed up by independent BnMST-genes deficient homozygous mutants.

The authors used computer programs to predict the subcellular localization of BnMST proteins. There is some profit in using of such programs but sometimes they  are  highly unreliable.  To better explore the localizations and potential interactions of the BnMST the protein localization should be performed in living plant cells  (e.g. transient expression in  Nicotiana leaves).

The manuscript is logically organized and written in an understandable English.

Could  you please add details on the experimental design, e.g. the number of replicates?

Generally, the data are well presented and the manuscript is easy to read. The authors did the good job, and presented enough comprehensive data regarding the topic of study. However, I would  like to advice to perform additional experiments on plants.  In my opinion this manuscript requires revision.

Author Response

(The authors gave the same response as above.)

Round 2

Reviewer 3 Report

I think that the authors provided a reasonable reply for my major points of concern.